# PRAGMA: A Foundation Model for Banking Event Sequences

Maxim Ostroukhov [1]   Ruslan Mikhailov [1]   Vladimir Iashin [1]   Artem Sokolov [1]   Andrei Akshonov [1]
Vitaly Protasov [1]   Dmitrii Beloborodov [1]   Vince Mullin [2]   Roman Enzmann [2]   Georgios Kolovos [2]   Jason Renders [2]
Pavel Nesterov [1]   Anton Repushko [1]

## Abstract

We present PRAGMA, a family of encoder-style foundation models for multi-source banking event sequences. PRAGMA is pre-trained with masked modelling on a large, heterogeneous corpus of user histories, using a key–value–time tokenisation and a two-branch encoder tailored to the discrete, variable-length nature of financial records. The pre-trained backbone transfers to a wide range of downstream tasks (fraud detection, product recommendation, lifetime value, and more), supporting both frozen-embedding probes and lightweight fine-tuning. Across six diverse banking benchmarks, PRAGMA matches or exceeds strong task-specific baselines from a single shared backbone, reducing the need for hand-crafted features. We report only relative improvements, as absolute metrics are commercially sensitive; all shown examples are synthetic.

## 1. Introduction

Foundation models trained at scale have transformed natural language processing (Devlin et al., 2019; Brown et al., 2020) and computer vision (Kirillov et al., 2023; Caron et al., 2021), yet their application to multi-source banking user histories remains comparatively underexplored. Banks and fintechs accumulate data of multiple source-types at scale: event streams of transactions, in-app navigation, and customer communications, alongside a raw static profile state capturing attributes such as account tenure and plan. Language models map poorly onto this setting: serialising structured records as text inflates sequence lengths, fragments numerical values, and discards both the irregular timestamp attached to each event and the natural key–value–time structure of a record. With no off-the-shelf architecture that jointly handles long-tailed, irregularly-timed, multi-source

histories, practitioners fall back on task-specific pipelines with extensive feature engineering.

Prior work addresses isolated slices of this problem. Tabular Transformers (Huang et al., 2020; Gorishniy et al., 2021) model fixed-schema rows. Financial language models (Yang et al., 2020; Wu et al., 2023; Yang et al., 2023) operate on text, while Time-LLM and Chronos (Jin et al., 2024; Ansari et al., 2024) tokenise numerical time series. Recent transaction models (Braithwaite et al., 2025; Dou et al., 2025) come closest, learning directly from transaction sequences, but they tokenise a single event type (transactions) and are evaluated on only a few tasks. In Braithwaite et al. (2025), additional user context is supplied as hand-crafted tabular features at fine-tuning rather than as raw fields learnt during pre-training. The literature still lacks a multi-source encoder backbone that learns raw task-agnostic profile state during pre-training and transfers across diverse banking tasks.

We present PRAGMA, a family of encoder-style foundation models for multi-source banking user histories. PRAGMA applies a key–value–time tokenisation with type-specific value encoding for numerical, categorical, and textual fields, and uses two encoder branches for profile state and events whose outputs are fused by a history encoder. We choose an encoder-only, bidirectional design for transferable discriminative representations, and adapt the pre-trained backbone via frozen embedding probes or LoRA (Hu et al., 2022). Across six banking benchmarks, a single pre-trained backbone outperforms strong task-specific baselines (Figure 1).

This work contributes: **1)** PRAGMA, a family of encoder-style foundation models for multi-source banking user histories, scaling from 10 M to 1 B parameters, combining key–value–time tokenisation with a two-branch profile/event design feeding a history encoder; **2)** an efficient pre-training recipe for long, irregular histories based on masked modelling, sequence packing, and dynamic batching; LoRA fine-tuning of the backbone consistently matches or exceeds training from scratch; **3)** an evaluation of a single backbone across six diverse downstream tasks (a substantially broader scope than prior models), including a failure mode on anti-money-laundering, where cross-record relational structure dominates.

---

[1]Revolut Research [2]NVIDIA. Correspondence to: Maxim Ostroukhov <first1.last1@revolut.com>.

*Proceedings of the $2^{nd}$ ICML Workshop on Foundation Models for Structured Data*, Seoul, South Korea. 2026. Copyright 2026 by the author(s).

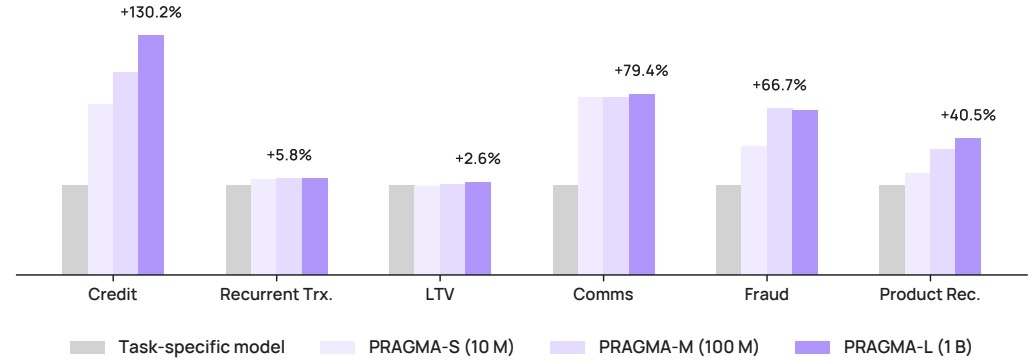

*Figure 1.* A single architecture from 10 M to 1 B parameters that outperforms task-specific models across tasks.

## 2. Method

### 2.1. Data

PRAGMA is pre-trained on a corpus of anonymised user records drawn from a large consumer fintech platform. Each record combines an ordered event history (transactions, in-app navigation, trading, and communications) with a static profile state describing time-invariant attributes (e.g., service region, balance quantile) and a small set of life-long events that carry the timestamp of milestone occurrences such as first top-up. The pre-training corpus covers 26 M user records, 24 B events, and 207 B tokens over a 25-month range (2023–2025); Section A gives a fuller description including Figure 4 and the rationale for the temporal window.

### 2.2. Tokenisation

Unlike standard LLMs that treat everything as text, a foundation model for banking events must preserve the structural nature and heterogeneity of tabular records. We represent each data point by three components: a semantic type (key), a value, and a temporal coordinate, as illustrated in Figure 2.

Keys are encoded as a single token, yielding a small vocabulary of 60 tokens shared across events and profile state. Values are encoded by type: numerical values are mapped to percentile buckets; low-cardinality values are treated as categorical values and encoded as a single token (e.g., a merchant category code); and high-cardinality textual values are tokenised with a BPE-style subword tokeniser (Sennrich et al., 2016), yielding ∼28 k value tokens in total.

Time is encoded in two ways. First, seconds to the most recent event, passed through a soft logarithmic transform $8 \cdot \ln(1+t/8)$. This compresses long ranges while preserving local granularity. Second, cyclical calendar features (hour of day, day of week, day of month) embedded with fixed-period sinusoids in the spirit of Gorishniy et al. (2022).

### 2.3. Architecture

PRAGMA is an encoder-only Transformer composed of three bidirectional blocks, illustrated in Figure 3. Key and value embeddings come from a shared lookup table and are summed with within-field sine/cosine positional embeddings; learnable [USR] and [EVT] tokens are prepended to each respective sub-sequence. A *Profile State Encoder* maps the profile-state tokens into a single [USR] embedding, using RoPE (Su et al., 2024) to encode log-seconds-to-life-long-event coordinates (zero for non-life-long attributes). An *Event Encoder* independently maps each event's tokens into an [EVT] embedding, to which calendar features are added. A *History Encoder* then contextualises the concatenated sequence of the profile token and event embeddings with log-seconds-to-last-event coordinates (again via RoPE), producing record-level representations used by the MLM head during pre-training and by downstream probes. PRAGMA is instantiated as a family of three variants: S (10 M params), M (100 M), and L (1 B), which jointly scale width, depth of each encoder, and the number of heads (see Section B and Table 2 for details).

**Pre-training objective.** We pre-train with a masked-language-modelling (MLM) objective following BERT (Devlin et al., 2019): for each masked token the MLM head receives the concatenation of the Event Encoder output at that token's position, the corresponding [EVT] position in the History Encoder, and the [USR] position in the History Encoder, projected back to $d$ dimensions and matched against the embedding table with a label-smoothed cross-entropy loss (Szegedy et al., 2016). Masking is applied to event tokens at three granularities: token-level (15 %), event-level (10 %, forcing whole event reconstruction), and semantic-type (10 %, masking all values for selected keys).

### 2.4. Downstream Adaptation

PRAGMA supports two adaptation modes. In the *embedding probe* mode, the frozen History Encoder output is fed

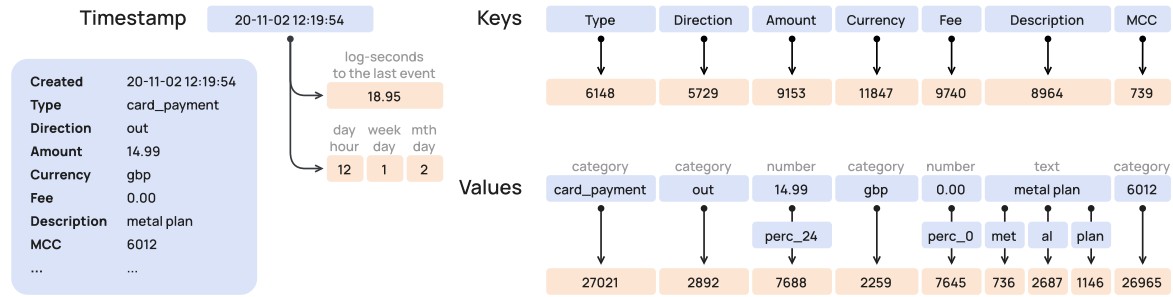

*Figure 2.* **Tokenisation overview.** A raw event record is decomposed into a timestamp, semantic types (keys), and values. Keys are single tokens; values use type-specific tokenisation (percentile buckets for numerical, single token for categorical, BPE subwords for textual). Time is encoded both as log-seconds to the last event and as cyclical calendar features. Profile state is encoded similarly to an event record.

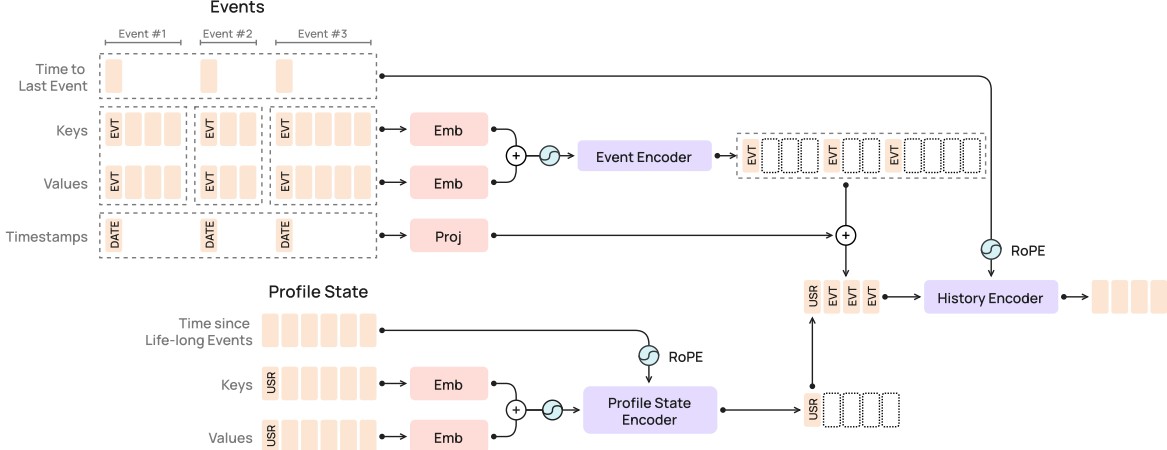

*Figure 3.* **PRAGMA backbone overview.** Each user record is represented as an ordered event history and profile state. A Profile State Encoder maps profile state into a `[USR]` embedding; an Event Encoder independently maps each event into an `[EVT]` embedding, augmented with calendar features. A History Encoder contextualises profile state and event history with time-to-last event coordinates, producing a record-level representation.

to a lightweight linear regression probe for rapid iteration. In the *LoRA* mode (Hu et al., 2022), low-rank updates are applied to the QKV and MLP projections of each encoder layer, updating only ∼2–4 % of parameters and keeping the pre-trained backbone largely frozen.

# 3. Experiments

For commercial-sensitivity reasons we do not report absolute downstream metrics and instead express all results as relative changes with respect to a task-specific reference.

**Tasks.** We evaluate PRAGMA on six internal downstream benchmarks spanning high-impact fintech use cases: *credit scoring* (binary classification); *communication engagement* (re-engagement of abandoned credit applicants, binary classification); *external fraud* (binary classification); *product recommendation* (multilabel adoption under communication treatments); *recurrent transactions* (binary recurrence detection); and *lifetime value* (binary positive-gross-profit classification). Each task has its own train/validation/test splits, and downstream datasets are assembled following

the same protocol as the pre-training corpus; full task and protocol details are deferred to Sections D and E.

## 3.1. Main Results

Table 1 (left) reports the headline result: pre-trained PRAGMA backbones, adapted with LoRA, consistently outperform task-specific baselines across all six benchmarks while sharing most parameters across tasks. At the largest scale, the biggest wins are on precision–recall metrics for high-stakes, low-prevalence targets: PR-AUC improves by +130.2 % on credit scoring and +79.4 % on communication engagement, suggesting the shared representation is particularly effective at identifying rare but high-value signals that traditional task-specific pipelines struggle to recover. ROC-AUC on the same tasks is more tempered but still substantial (+12.4 % and +20.4 %), external fraud moves by +16.7 %/ + 64.7 % on precision/recall, and product recommendation gains +40.5 % mAP. Gains on lifetime value and recurrent transactions are smaller (+1.8 to +5.8 %), consistent with tasks where simple aggregations of recent activity already capture most of the signal.

| Task | Metric | Baseline | PRAGMA | | | S | | M | | L | |
|---|---|---|---|---|---|---|---|---|---|---|---|
| | | | S | M | L | Emb | LoRA | Emb | LoRA | Emb | LoRA |
| Credit scoring | PR-AUC | – | +70.3 | +98.0 | +130.2 | – | +18.0 | – | +20.4 | – | +10.3 |
| | ROC-AUC | – | +6.2 | +10.1 | +12.4 | – | +0.2 | – | +2.4 | – | +1.5 |
| Comm. engagement | PR-AUC | – | +76.6 | +76.7 | +79.4 | – | +72.9 | – | +49.7 | – | +54.1 |
| | ROC-AUC | – | +19.6 | +17.4 | +20.4 | – | +16.9 | – | +11.2 | – | +13.5 |
| External fraud | Precision | – | +0.3 | +12.3 | +16.7 | – | +30.8 | – | +29.8 | – | +23.8 |
| | Recall | – | +33.4 | +66.4 | +64.7 | – | +27.4 | – | +24.5 | – | +13.3 |
| Product rec. | mAP | – | +10.6 | +31.5 | +40.5 | – | +57.2 | – | +68.4 | – | +68.1 |
| Recurrent txns | $F_1$ | – | +5.4 | +6.0 | +5.8 | – | +4.5 | – | +3.2 | – | +2.3 |
| Lifetime value | PR-AUC | – | −1.2 | +0.3 | +1.8 | – | +3.6 | – | +2.4 | – | +2.9 |
| | ROC-AUC | – | −0.8 | +0.9 | +2.6 | – | +4.7 | – | +3.4 | – | +3.9 |

*Table 1.* **Left: LoRA-adapted PRAGMA backbones outperform task-specific baselines across all six downstream tasks; gains generally grow with scale. Right: within each model size, LoRA fine-tuning consistently outperforms embedding-only probes on the same backbone.** *Left:* for each model size (S/M/L), relative performance is computed as (PRAGMA/Baseline − 1)%. *Right:* within each model size, the embedding-only variant (Emb) is the reference; gains are computed as (LoRA/Emb − 1)%.

## 3.2. Effect of Scale

Scaling behaviour is task-dependent (Table 1, left). Credit scoring, product recommendation, and external fraud reward scale (e.g. credit-scoring PR-AUC grows from +70.3 % at PRAGMA-S to +130.2 % at PRAGMA-L), whereas communication engagement, lifetime value, and recurrent transactions are largely small-model tasks where PRAGMA-S already lands within a few points of the largest backbone. This points to an efficiency sweet spot in which light backbones suffice for stable, behaviour-driven targets, while only harder discriminative tasks continue to reward scale.

## 3.3. LoRA vs. Embedding Probes

Table 1 (right) isolates the contribution of the adaptation head: across all tasks and scales, LoRA-tuned variants consistently outperform embedding-only probes on the same backbone, confirming that the gains in the headline result come from the pre-trained representation jointly with parameter-efficient adaptation rather than from frozen embeddings alone. The pattern is broadly stable across scales: even the smallest backbone benefits substantially from LoRA, suggesting that frozen embeddings under-utilise capacity already present at PRAGMA-S and that parameter-efficient adaptation is a cheaper lever than larger frozen features. See Section F for additional ablations.

## 3.4. Effect of Pre-training

Pre-training is load-bearing (Table 3). Comparing LoRA fine-tuning of the pre-trained backbone against task-specific training from scratch at the same 100 M scale, LoRA matches or exceeds scratch on every evaluated task: +18.6 % PR-AUC on communication engagement, +13.0 % on credit scoring, and +10.3 % mAP on product recommendation, with parity on the saturating tasks. This indicates

that the masked-modelling objective recovers task-relevant structure that a single-task supervised loss does not, even when the scratch model is given access to the same architecture and parameter budget.

## 3.5. Limitations: Anti-Money Laundering

Not every fintech task benefits equally. Formulating anti-money laundering (AML) as a binary classification problem and probing PRAGMA-L embeddings with a linear head yields a −47.1 % relative drop in $F_{0.5}$ against an internal baseline. We attribute this primarily to AML being inherently relational: the baseline leverages cross-record features that capture network-level signals, whereas PRAGMA processes each user history in isolation and therefore cannot represent such dependencies. This delineates a boundary for the current paradigm and points to extending the backbone with cross-record context as a concrete future direction.

## 4. Conclusion

We presented PRAGMA, a family of encoder-style foundation models for multi-source fintech user histories that combines a key–value–time tokenisation with a two-branch architecture fused by a history encoder, pre-trained with masked modelling on 207 B tokens of heterogeneous fintech events. Across six diverse downstream tasks spanning fraud detection, lifetime value, and credit scoring, among others, a single pre-trained backbone outperforms strong task-specific baselines directly from raw event sequences, with LoRA fine-tuning matching or exceeding training from scratch while updating only a small fraction of parameters. The AML case study delineates a clear limitation: tasks that depend on cross-record relational structure remain out of reach for a model that processes event histories in isolation, which we regard as a concrete direction for future work.

**Acknowledgments**    We thank Mathilde Horanieh, Dmitry Mittov, Ian Iakobsen, Aleksandr Pushin, Muhammad Anas, Viacheslav Karpov, Nathalie Skrzypek, Leyla Sultanova, Francisco Sanz Estevez, Nikita Kravchuk, Tadas Krisciunas, Amey Baokar, Hanna Danilovich, Jyoti Prakash Bal, Vitalii Radchenko, Kade Main, Nic Hatia, and other Revoluters for their contributions to this work.

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

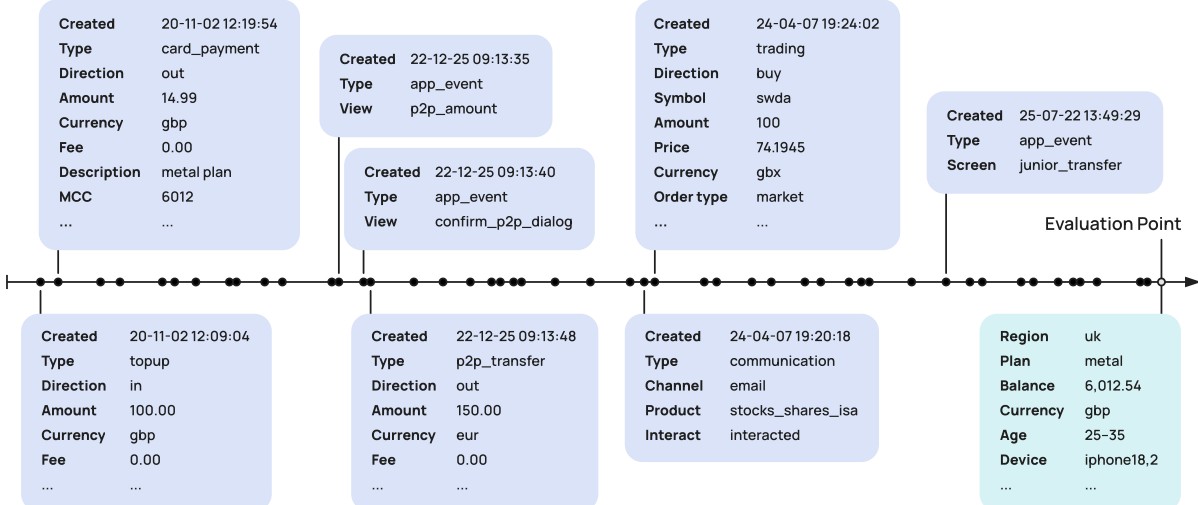

*Figure 4.* **Event timeline overview.** After account creation, users generate a sequence of platform interactions over time, spanning transactions, in-app navigation, and communications. We aggregate the event history up until a designated evaluation point. Alongside these sequential events, we capture contextual attributes that describe the record's state at that point, e.g., membership plan or service region. Both events and attributes share a uniform representation: a timestamp and a set of key–value pairs (e.g., `Type: card_payment`, `Channel: email`). All values shown are synthetic; the figure is for illustration purposes only.

## A. Dataset Detail

Our goal is to build a foundation model that encodes diverse event-level signals and transfers across a wide range of downstream tasks. The dataset is structured at the record level, where each observation represents a pseudonymised event history associated with an evaluation point. As shown in Figure 4, we consider an event history alongside contextual attributes. This lets the model account for both sequential patterns and time-invariant features like account currency. All data used in this work is fully anonymised and contains no personally identifiable information. We construct our pre-training dataset from 26 M user records spanning 111 countries, accumulating 24 B events that total 207 B tokens.

**Event History.** Standard platform usage generates event streams across various services, e.g., account funding, payments, in-app navigation, or service communications. These aggregated event histories capture population-level patterns that support a range of analytical and predictive tasks. An event is defined by a created timestamp and a set of key–value pairs, e.g., `Direction: out`. We fetch events from broad source types loosely grouped into transactions, app, trading, and communication, selected for their high expected impact on downstream tasks. Event schemas are specific to their source type and incorporate distinct sets of keys, e.g., `Symbol` is unique to trading events. Beyond anonymisation, de-identification, and standard eligibility criteria, no additional statistical filtering or pre-processing (e.g., outlier removal or vocabulary pruning) is applied to the event streams, ensuring that the model captures the full heterogeneity found in production.

**Profile State.** Alongside the event history we incorporate general contextual attributes such as balance quantile, plan, insurance state, and service region. These attributes provide useful context otherwise missing from the event history. Profile state is a set of descriptive key–value pairs in an event-like format, e.g., `Plan: metal`, timestamped at the designated evaluation point (or the cut-off date during pre-training).

High-activity users often generate tens of thousands of interactions, exceeding computational bounds; we address this via truncation to a fixed context window (Section C). However, truncation risks discarding early historical milestones that carry useful signal, such as account age. We therefore augment profile state with *life-long events*, key–value pairs that, unlike regular profile attributes, each carry an individual timestamp recording a first occurrence, e.g., `Lifelong: first_topup` at `20-11-02 12:09:04`. This timestamp is then used to compute the temporal distance to the evaluation point, enabling the model to encode the timing of historical milestones.

**Pre-training Time Range.** Developing a robust and generalisable model requires a delicate balance between maximising historical coverage and maintaining data relevance. Older events may reflect historical patterns, product features, or system

dynamics that are no longer relevant at inference time, creating a distribution mismatch that can degrade performance. The inclusion of highly heterogeneous events from long time spans can also make the pre-training task harder and slow down convergence. Conversely, downstream applications may require predictions on events much earlier or later than those used for pre-training, and Transformer architectures have a limited effective context span. With these considerations in mind, we select a temporal range of 25 months from 2023 to 2025 for pre-training, balancing comprehensive event coverage, recency, distribution consistency, and tractable sequence modelling.

## B. Architecture Detail

PRAGMA is an encoder-only Transformer that inputs an event history along with contextual attributes and outputs dense record-level embeddings. It is trained on a large-scale, diverse dataset with a masked modelling (MLM) objective that reconstructs masked input tokens. Once pre-trained, it acts as a backbone for downstream adaptation with small-scale ($\sim$2–4 %; of the model's parameters) fine-tuning for a variety of tasks. The high-level overview is shown in Figure 3. PRAGMA is parametrised as a family of models with 10 M, 100 M, and 1 B parameters, enabling selection according to operational budget and constraints; details are in Table 2. All variants use GELU activations (Hendrycks & Gimpel, 2016), pre-norm layer normalisation (Xiong et al., 2020), and dropout of 0.1 (Srivastava et al., 2014).

| Model | Params | Width | | Depth | | | Heads |
| --- | --- | --- | --- | --- | --- | --- | --- |
| | | $d_{\mathrm{model}}$ | $d_{\mathrm{ffn}}$ | Profile | Event | History | |
| PRAGMA-S | 10 M | 192 | 768 | 1 | 5 | 2 | 3 |
| PRAGMA-M | 100 M | 512 | 2048 | 3 | 16 | 6 | 8 |
| PRAGMA-L | 1 B | 1024 | 4096 | 9 | 45 | 18 | 16 |

*Table 2.* **PRAGMA model family.** PRAGMA scales across three variants by jointly increasing model width ($d_{\mathrm{model}}$, $d_{\mathrm{ffn}}$), depth of the profile-state, event, and history encoders, and the number of attention heads.

The model consists of three main blocks: Profile State Encoder, Event Encoder, and History Encoder. First, the profile state tokens are processed by the Profile State Encoder. Second, similar to profile state, each event is encoded independently in the Event Encoder. Finally, the outputs of the Profile State and Event Encoders are concatenated and encoded in the History Encoder to form an output. Depending on the stage, the final output is used either in an MLM head during pre-training, a classification head during fine-tuning, or as-is in an embedding probe.

**Token Embedding.** Profile state and event tokens are embedded identically. For multi-valued fields (e.g., `Description`), the key token is replicated to match each of its values, yielding $n$ key–value pairs in total. A single shared embedding table $E$ maps each key and value to a $d$-dimensional vector; the two embeddings are summed and augmented with static sine/cosine positional encodings (PosEmb) (Vaswani et al., 2017):

$$x = \mathrm{PosEmb}\big(E(k) + E(v)\big), \quad x \in \mathbb{R}^{n \times d}. \tag{1}$$

Positions index values *within* a field, not across fields; e.g., the value `eur` of `Currency` receives position 0, while the three value tokens (`met`, `al`, `plan`) of `Description` receive positions (0, 1, 2) (see Figure 2). We denote user and event embeddings as $x_a \in \mathbb{R}^{n_a \times d}$ and $x_e \in \mathbb{R}^{n_e \times d}$, respectively. Following common practice in encoder-only Transformers (Devlin et al., 2019; Dosovitskiy et al., 2021), a learnable [USR] (or [EVT]) token is prepended to each sequence (Figure 3).

**Profile State Encoder.** The Profile State Encoder is a bidirectional Transformer. It inputs the profile state tokens $x_a \in \mathbb{R}^{n_a \times d}$ and corresponding temporal coordinates $t_a \in \mathbb{R}^{n_a}$, where each entry holds the log-seconds since the corresponding life-long event (or 0 for non-life-long pairs). We use RoPE (Su et al., 2024) to encode $t_a$, disentangled from the value-level positional embedding above to avoid semantic and scale mismatch. The output is a sequence of profile-state embeddings $z_a \in \mathbb{R}^{n_a \times d}$; we pass the first element, corresponding to the [USR] token, to the History Encoder (denoted $z_a \in \mathbb{R}^{1 \times d}$ for simplicity).

**Event Encoder.** The Event Encoder is a bidirectional Transformer similar to the Profile State Encoder. It inputs an event history $x_e = (x_{e,1}, x_{e,2}, \ldots, x_{e,n_e})$, where each element has a distinct number of token embeddings ($x_{e,i} \in \mathbb{R}^{n_i \times d}$), and

processes each event independently of all others. The module outputs a token-level embedding sequence for each event, denoted $\widehat{z}_e$, used by the MLM head during pre-training. The first token corresponding to [EVT] is selected as the aggregated representation $z'_e \in \mathbb{R}^{n_e \times d}$. Calendar features (hour of day, day of week, day of month) $x_t \in \mathbb{R}^{n_e \times 3}$ are converted to sine and cosine radians and embedded with two MLP layers into $z_t \in \mathbb{R}^{n_e \times d}$, then added to the Event Encoder output: $z_e = z'_e + z_t$.

**History Encoder.** The History Encoder is a bidirectional Transformer similar to the other two encoders. It inputs the concatenated aggregated representations $z = [z_a : z_e] \in \mathbb{R}^{(1+n_e) \times d}$ and the corresponding temporal coordinate $t_e \in \mathbb{R}^{1+n_e}$, where each entry holds the log-seconds to the most recent event (0 for the $z_a$ position), again encoded via RoPE. The output is a sequence of embeddings $z_h \in \mathbb{R}^{(1+n_e) \times d}$, where $z_{h,0}$ corresponds to [USR] and $z_{h,1}, \ldots, z_{h,n_e}$ to the [EVT] tokens. $z_h$ is used by the MLM head during pre-training and for downstream probes.

**Pre-training Objective.** PRAGMA is pre-trained with an MLM objective following BERT (Devlin et al., 2019): a random subset of event input tokens is masked and the model reconstructs the original tokens. For each masked token, the MLM head receives the concatenation of three $d$-dimensional vectors: the Event Encoder output at that token's position within $\widehat{z}_e$ (local within-event context); the History Encoder output at the corresponding [EVT] position $z_{h,i}$ (cross-event context); and the History Encoder output at the [USR] position $z_{h,0}$ (user-level context). This $3d$-dimensional representation is projected back to $d$ and matched against the embedding table to produce logits; the training loss is cross-entropy with label smoothing (Szegedy et al., 2016).

**Masking Strategy.** The masking strategy combines three sources: individual token-level masking (with 15 % probability), event-level masking (10 %) that requires reconstructing an entire event, and semantic-type (key)-level masking (10 %) where all values of selected keys are masked, training the model to predict values given context and a key. A small fraction of selected positions are replaced with [UNK] rather than [MASK]; [UNK] positions are excluded from the MLM objective and therefore receive no gradient, acting as a form of input dropout and training the model to recover original values under a stronger corruption scheme.

# C. Training Infrastructure

Pre-training PRAGMA on 207 B tokens spanning 24 B user events introduces several engineering challenges. The heterogeneous, table-structured nature of the data requires specialised storage, batching, and truncation strategies; each is described below.

**Data Storage.** The pre-training corpus is stored as a two-level structure: a *user index* (an LMDB-backed key-value store mapping each user to their tokenised profile state and per-user token statistics) and a collection of *event shards* (Parquet files partitioned by event count, so each file contains only users with the same number of events). This layout lets workers stream event shards independently and look up profile state on demand.

**Batching.** Each training sample consists of a complete event history together with its associated profile state tokens. Because event histories vary greatly in length, from a handful of events to thousands, naïve padding-based batching would waste most compute on padding tokens. Sharding records by event count avoids many random-access disk operations during loading and yields uniform-length event sequences within each batch, so the History Encoder operates on a rectangular tensor without ragged or padded dimensions. We employ *dynamic batching* with a fixed token budget that fits into GPU memory: records from the same shard are greedily packed until the budget is reached.

**Sequence Packing.** Within a batch, individual events still vary in their number of tokens. Rather than padding every event to the longest one, we pack all event tokens into a flat buffer and process them with a variable-length (varlen) attention kernel (Dao et al., 2022), so tokens from different events do not attend to each other at this stage. Together with shard-based batching, this eliminates padding overhead along both the event and token axes. Compared to a padded baseline, sequence packing coupled with dynamic batching yields a 2–5× throughput improvement depending on the sequence-length distribution.

**Truncation.** To bound memory consumption at a fixed context length, we apply two levels of truncation before packing. At the *event level*, each individual event is truncated to at most 24 tokens, affecting only 0.01 % of events. At the *profile-state*

*level*, the static profile-state sequence is truncated to at most 200 tokens. Users with zero events are discarded; users with more than 6,500 events are subsampled by retaining the most recent ones, preserving temporal recency.

**Pre-training Compute.** The three model variants were trained with bf16 mixed precision and the Muon optimiser combined with AdamW (Loshchilov & Hutter, 2019; Jordan, 2024; Liu et al., 2025). PRAGMA-S (10 M) and PRAGMA-M (100 M) were trained on $16\times$ H100 GPUs, and PRAGMA-L (1 B) on $32\times$ H100 GPUs. The smallest variant converged in approximately two days, while the 100 M and 1 B models each required roughly two weeks of wall-clock time.

## D. Evaluation Protocol

We evaluate PRAGMA primarily via embedding probes and Low-Rank Adaptation (LoRA) (Hu et al., 2022) fine-tuning on downstream tasks.

**Embedding Probing.** Embedding probing facilitates rapid iteration during experimentation before committing to LoRA fine-tuning, e.g., to gauge whether a new feature brings the expected gain, to select a checkpoint after a pre-training run, or to decide whether to explore a task as a downstream target at all. Embeddings are extracted from the History Encoder output ($z_h$). For probing we evaluate the [USR] token, the final [EVT] token, and their concatenation, using a standard linear probe. Given predefined train/validation/test partitions, we forward each record through the frozen encoder to obtain fixed-size representations and train a linear probe (logistic or linear regression). Probe performance is robust to hyper-parameters, so fitting typically takes a couple of minutes; embeddings are standard-scaled prior to fitting, and the L-BFGS optimiser (Liu & Nocedal, 1989) yields the best results and converges quickly. Gradient Boosted Decision Trees can perform well on lower-dimensional embeddings (e.g., 192-d), but their need for per-task hyper-parameter tuning and slower fit time make them less practical than linear probing for high-velocity model evaluation.

**Downstream Adaptation with LoRA.** To specialise the PRAGMA backbone for downstream tasks, we employ LoRA, which introduces a minimal parameter overhead of only 2–4 %. The pre-trained weights are fine-tuned for task-specific objectives, bridging the gap between general representation learning and downstream requirements. We apply LoRA to QKV projections and MLP layers within encoder layers, following common practice (Hu et al., 2022; Dettmers et al., 2023), and default to rank $= 8$ with $\alpha = 8$ across all experiments, also sweeping the rank across $\{4, 8, 16\}$ on smaller datasets. We use Adam (Kingma & Ba, 2015) for LoRA fine-tuning, and training typically uses $1/8$ of the wall-clock time of pre-training, converging in 12 hours to a few days depending on dataset size.

**Preparing Downstream Datasets.** For each downstream task, we obtain a unique identifier, typically a profile id and an evaluation point. We gather the event history and profile attributes directly preceding the evaluation point and follow the pre-defined folds and splits for each downstream task. The downstream dataset collection process mirrors that of the pre-training dataset.

## E. Downstream Tasks

**Credit Scoring.** The task is to assess credit risk for retail applications by predicting the probability of default within the first 12 months of use. The downstream dataset spans multiple years and is diverse across records. This is a binary classification problem with a minority class; performance is measured with ROC-AUC and PR-AUC offline metrics.

**Communication Engagement.** The task is to predict whether a user who abandoned a credit application mid-process will open a re-engagement communication. This action serves as an upper-funnel proxy for resuming the application and eventually originating a loan. A distinguishing aspect is the severely limited sample size, requiring the model to capture nuanced event-level signals from minimal data. This is a binary classification problem, with ROC-AUC and PR-AUC as the main metrics.

**External Fraud.** A representative fraud-detection use case formulated as a binary classification problem, evaluated with precision and recall.

**Product Recommendation.** The task is to predict which products a user is likely to adopt in the near future, conditioned on receiving a specific communication (e.g., email or push notification). A key challenge is modelling conversion propensity

| Task | Metric | PRAGMA-M | |
| | | Scratch (ref.) | LoRA |
|---|---|---|---|
| Comm. engagement | PR-AUC | – | +18.6 % |
| | ROC-AUC | – | +5.0 % |
| Credit scoring | PR-AUC | – | +13.0 % |
| | ROC-AUC | – | +1.6 % |
| External Fraud | Precision | – | +8.0 % |
| | Recall | – | +17.0 % |
| Product rec. | mAP | – | +10.3 % |
| Recurrent txns | $F_1$ | – | +0.6 % |
| Lifetime value | PR-AUC | – | +0.4 % |
| | ROC-AUC | – | +0.3 % |

*Table 3.* **LoRA fine-tuning consistently matches or exceeds task-specific training from scratch.** Relative performance computed as (LoRA/Scratch − 1).

across multiple products simultaneously while accounting for the contextual influence of the communication. Formulated as a multilabel classification problem; the main metric is mean average precision (mAP).

**Recurrent Transactions.** This task predicts whether a given transaction corresponds to a recurring subscription that will repeat in the following month. A key challenge is distinguishing true recurring patterns from irregular or one-off payments given limited historical signals. Formulated as a binary classification task, evaluated with macro-averaged $F_1$-score to account for class imbalance.

**Lifetime Value (LTV).** The LTV task assesses the probability of a user generating positive gross profit, formulated as a binary classification problem. A distinguishing aspect is that users have shorter event histories (e.g., a couple of weeks), while the prediction horizon is typically six months or more. Main metrics are ROC-AUC and PR-AUC.

## F. Ablations

### F.1. Effect of Pre-training (LoRA vs Scratch)

Table 3 shows that LoRA fine-tuning consistently matches or exceeds full-parameter training from scratch across all evaluated tasks. The largest gains are on Communication Engagement (+18.6 % PR-AUC, +5.0 % ROC-AUC), suggesting that the pre-trained backbone captures rich, diverse event patterns that are difficult to learn when training from scratch on a single downstream task. Credit Scoring and Product Recommendation show similar patterns (+13.0 % PR-AUC, +1.6 % ROC-AUC, and +10.3 % mAP respectively). For Recurrent Transactions and Lifetime Value, improvements are more modest, indicating that the scratch-trained baselines already capture most of the task-relevant structure and LoRA fine-tuning maintains parity without regression. These findings are particularly significant for production environments, as they confirm that PRAGMA can consolidate multiple independent, high-maintenance models into a single shared system without sacrificing predictive accuracy while maintaining a significantly smaller trainable parameter footprint.

### F.2. Effect of Profile State

Table 4 isolates the contribution of the Profile State Encoder by comparing the full PRAGMA-S against a variant that removes the profile-state branch, relying solely on event-level representations. The impact is strongly task-dependent. Credit Scoring benefits substantially, with a +31.8 % relative gain in PR-AUC and +4.9 % in ROC-AUC; the outsized PR-AUC improvement indicates that profile state is particularly valuable for identifying the minority default class, where static signals such as account tenure and onboarding characteristics provide discriminative context that event sequences alone cannot fully capture. External fraud sees even larger precision/recall gains (+46.8 %/ + 85.6 %) for the Small model. By contrast, Lifetime Value shows more moderate gains of +2.2 % in PR-AUC and +2.0 % in ROC-AUC, suggesting that gross-profit

| Task | Metric | PRAGMA-S | |
|---|---|---|---|
| | | Event-only (ref.) | Full |
| External fraud | Precision | – | +46.8 % |
| | Recall | – | +85.6 % |
| Credit scoring | PR-AUC | – | +31.8 % |
| | ROC-AUC | – | +4.9 % |
| Product rec. | mAP | – | +3.5 % |
| Lifetime value | PR-AUC | – | +2.2 % |
| | ROC-AUC | – | +2.0 % |
| Recurrent txns | $F_1$ | – | +2.4 % |
| Comm. engagement | PR-AUC | – | −3.0 % |
| | ROC-AUC | – | +1.3 % |

*Table 4.* **Profile state contributes substantially to tasks where static user characteristics are discriminative.** Relative performance is computed as (Full/Event-only − 1).

likelihood is largely inferable from transactional patterns alone. Communication Engagement exhibits a slight PR-AUC regression (−3.0 %) alongside a marginal ROC-AUC gain (+1.3 %), indicating that re-engagement propensity is driven almost entirely by pre-drop-off event patterns rather than static user characteristics. These results validate the two-branch design: the dedicated Profile State Encoder adds significant value for tasks where static profile state is informative, while the architecture degrades gracefully when those signals are less relevant.

### F.3. Communication Engagement (Uplift)

This task moves beyond conversion prediction to optimal treatment selection: the goal is to identify which messaging strategy best re-engages users with abandoned credit applications. The dataset is smaller in scale than our other downstream benchmarks, yet large-scale pre-training proves decisive, significantly outperforming a baseline trained on the limited in-domain data alone. As an uplift task, it also offers a distinct evaluation angle: PRAGMA is used as a frozen feature extractor feeding a meta-learner rather than being fine-tuned, isolating representational quality in the absence of task-specific adaptation. Concretely, we adopt a meta-learner framework (Künzel et al., 2019) to estimate heterogeneous treatment effects. Both PRAGMA and the baseline use the same meta-learner, differing only in the underlying representation. Table 5 reports results using Area Under the Uplift Curve (AUUC) and SNIPS (Swaminathan & Joachims, 2015). PRAGMA-L achieves a relative AUUC increase of +163.7 % over the internal baseline, translating its ability to capture latent event-level patterns into highly effective treatment allocation.

| Task | Metric | Baseline (ref.) | PRAGMA |
|---|---|---|---|
| Comm. engagement (uplift) | AUUC | – | +163.7 % |
| | SNIPS | – | +10.8 % |

*Table 5.* **PRAGMA-L vs. the internal uplift baseline under the same meta-learner framework.** Relative performance is (PRAGMA-L/Baseline − 1).

### F.4. Effect of a Pre-trained Text Encoder

In the standard PRAGMA architecture, text values are learned jointly with all other tabular features via an embedding lookup table (see Section B). To prevent the model from underfitting sparse, noisy, or highly irregular financial text (e.g., truncated transaction descriptions), we investigate offloading text comprehension to a dedicated, pre-trained text embedding model, e.g., Nemotron-1B-v2 (de Souza P. Moreira et al., 2024). This decoupled approach provides richer, out-of-the-box semantics and frees the primary Event Encoder (Section B) to focus on cross-feature interactions. While we do not use this as the default formulation in our generalised core architecture, we report on it as an optional extension that offers valuable domain-specific insights.

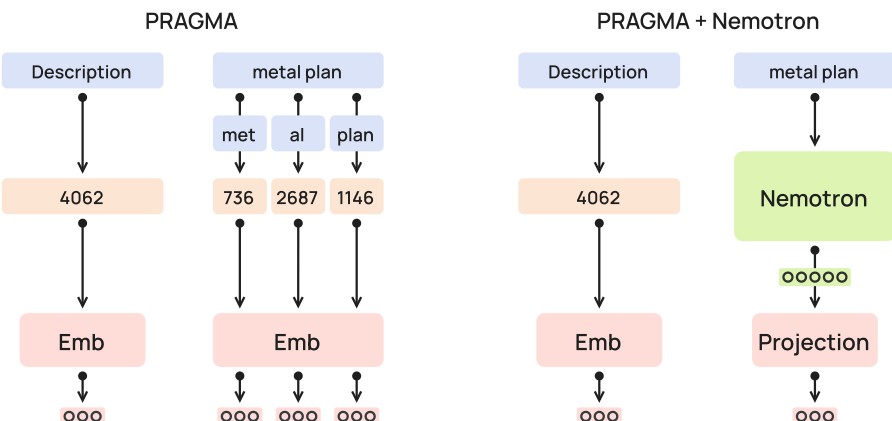

*Figure 5.* **Text embedding with PRAGMA (left) compared to a version with pre-trained Nemotron-1B-v2 text embedding (right).** Instead of our custom trained BPE tokeniser and a trainable embedding lookup table, a pre-trained "frozen" Nemotron maps an entire text value to a single text embedding vector which is projected into the Transformer's base dimension with a trainable projection.

| | | PRAGMA-M | |
|---|---|---|---|
| **Task** | **Metric** | **ref.** | **+Nemotron** |
| Credit scoring | PR-AUC | – | +16.1 % |
| | ROC-AUC | – | +2.8 % |
| Recurrent txns | $F_1$ | – | +0.1 % |
| Lifetime value | PR-AUC | – | +0.8 % |
| | ROC-AUC | – | +0.6 % |
| External fraud | Precision | – | +3.8 % |
| | Recall | – | −0.7 % |
| Product rec. | mAP | – | −6.4 % |

*Table 6.* **Impact of pre-trained text embeddings on downstream tasks is concentrated in text-heavy domains.** Performance is reported relative to a LoRA-tuned PRAGMA-M.

**Implementation Details.** The addition of a pre-trained text encoder involves multiple structural changes to the PRAGMA architecture. First, for semantic types (keys) whose values are normally encoded using a custom-trained BPE tokeniser and a trainable embedding lookup table, we instead use the frozen pre-trained model to map the complete text string to a single vector, which is then adapted via a one-layer trainable projection (see Figure 5). Second, instead of reconstructing exact token labels for these text fields during MLM optimisation, we train PRAGMA to reconstruct the continuous text embedding produced by the pre-trained text encoder with Mean Squared Error (MSE).

**Results & Discussion.** The results are shown in Table 6. Downstream effects track how much label-relevant signal sits in free text versus categorical and behavioural structure. Credit Scoring shows the clearest upside, with +16.1 % relative PR-AUC and +2.8 % ROC-AUC under Nemotron. Product Recommendation instead loses ground: mAP drops by −6.4 % relative, plausibly because sparse text adds little beyond what the structural channels already encode. External Fraud moves modestly and in opposite directions on precision (+3.8 %) versus recall (−0.7 %), while LTV and Recurrent Transactions stay near flat on the reported metrics. Because this variant also increases PRAGMA-M training latency by about 18 %, we keep it as an opt-in module for text-heavy tasks rather than baking it into the default architecture.

