# OpenReview forum: "PRAGMA: A Foundation Model for Banking Event Sequences"
_ICML.cc/2026/Workshop/FMSD — FMSD @ ICML 2026 Poster_

### Official Review · Reviewer_vrGM · 2026-05-19
**Promising banking event foundation model, but empirical transparency is limited**

**Rating:** 5
**Confidence:** 4

**Review:**

**Summary**

This paper introduces PRAGMA, an encoder-style foundation model for multi-source banking event sequences. It uses a key–value–time tokenization scheme, separate encoders for profile state and event histories, and a history encoder to produce transferable user representations. The model is pre-trained on a large proprietary fintech corpus and evaluated on six internal downstream tasks.

Overall, the paper is relevant, practical, and technically interesting. However, the empirical evidence is difficult to fully assess because results are reported only as relative improvements over internal baselines, with limited information about absolute performance, baseline strength, dataset sizes, class prevalence, uncertainty, and leakage controls.

**Strengths**

The problem is important and well aligned with the workshop theme. Banking event histories are heterogeneous, irregularly timed, and multi-source, making them a natural target for structured-data foundation models.

The architecture is well motivated. The key–value–time representation and the separation between profile-state, event-level, and history-level encoding fit the structure of the data.

The scale of the work is impressive, with pre-training over a very large fintech corpus.

The paper evaluates the model on several practical downstream tasks, including credit scoring, fraud detection, product recommendation, recurrent transactions, communication engagement, and lifetime value.

The paper includes useful ablations, such as LoRA vs. embedding probes, pre-training vs. scratch, the role of profile state, and the use of an external text encoder.

**Areas for Improvement**

The main weakness is the lack of absolute metrics. Reporting only relative improvements makes it hard to judge whether the gains are practically meaningful.

The internal baselines are not described in enough detail. Since all results are relative to these baselines, the paper should explain what models they are, what features they use, and how strongly they were tuned.

The evaluation is entirely internal and proprietary, which limits reproducibility. More anonymized aggregate information would help, such as dataset sizes, class prevalence, split dates, and number of records per task.

Potential leakage is not sufficiently discussed. Since pre-training and downstream evaluation come from the same platform and overlapping time range, the authors should clarify whether future events or test users appear in pre-training.

For high-stakes tasks such as credit scoring and fraud detection, ROC-AUC and PR-AUC are not enough. Calibration, thresholded performance, fairness, and segment-level robustness should also be discussed.



**Detailed Comments**


Please report absolute or normalized absolute metrics in addition to relative improvements. Large relative gains can be hard to interpret without knowing the baseline level.
Please describe the task-specific baselines more clearly. Are they production models, gradient-boosted trees, deep models, or hand-engineered pipelines?
Please clarify leakage controls: whether downstream test users appear in pre-training, whether pre-training includes events after the downstream evaluation point, and whether splits are temporal or random.
Please report dataset sizes, class prevalence, time ranges, and train/validation/test split sizes for each downstream task.
Please include confidence intervals or bootstrap estimates for the reported improvements.
For credit and fraud tasks, please discuss calibration, threshold-level performance, and robustness across user segments.
Please clarify the scope of the foundation-model claim. The model transfers across internal tasks, but it is not clear whether it transfers across institutions, geographies, products, or shifted time periods.
The scaling analysis should be expanded. Some tasks appear to saturate with smaller models, so the paper should clarify when the 1B model is worth the extra cost.


**Justification of Score**

I would assign this paper a 5: marginally below acceptance threshold.

The paper is ambitious, relevant, and practically valuable. The architecture is sensible, the scale is impressive, and the evaluation covers several important banking tasks. However, the current evidence is too opaque to fully support the claims. The lack of absolute metrics, limited baseline details, missing uncertainty estimates, and unclear leakage controls make it difficult to judge the true strength of the results.

Overall, I view PRAGMA as a promising industrial foundation-model system for banking event sequences, but the paper needs more transparent and rigorous empirical reporting before its claims can be fully assessed.

---

### Official Review · Reviewer_1164 · 2026-05-20
**PRAGMA: A Foundation Model for Banking Event Sequences**

**Rating:** 9
**Confidence:** 4

**Review:**

The paper describes how to generate embeddings for banking events by pretraining with MLM and
subsequently fine-tuning for specific tasks, such as predicting the probability of credit default,
detecting fraud, etc. The data consists of structured banking events (e.g., withdrawals, deposits, transfers)
and static account profiles. Three bidirectional encoders are used to generate the embeddings.
The authors demonstrate that PRAGMA typically outperforms task-specific models on six different benchmarks.

The paper is highly relevant to this workshop, as it applies foundation models -- bidirectional transformer encoder -- to structured banking time-series data. The authors demonstrate that this approach typically outperforms traditional task-specific models and therefore others can benefit from it.

On the one hand, the authors state that they use synthetic data; on the other hand, they report only relative performance improvements over the baseline, citing "commercial sensitivity" as justification. If the data is indeed synthetic, commercial sensitivity concerns should no longer apply. Additionally, it would be valuable to publish this synthetic dataset to enable reproducibility.

---

### Official Review · Reviewer_p6Cb · 2026-05-22
**PRAGMA: A Foundation Model for Banking Event Sequences**

**Rating:** 5
**Confidence:** 5

**Review:**

## Summary
This paper presents PRAGMA, a family of encoder-only foundation models (10M / 100M / 1B parameters) for multi-source banking user histories. Each user record is represented as an ordered event history (transactions, in-app navigation, communications) paired with a static profile state (balance quantile, service region, plan, life-long milestone events). The tokenisation decomposes each event into key-value-time triples: keys map to a shared 60-token vocabulary; numerical values are bucketed into percentiles, categorical values are single tokens, and high-cardinality text uses BPE (~28k value tokens total); time is encoded both as log-seconds since last event (soft logarithmic transform 8*ln(1+t/8)) and as cyclical calendar features via fixed-period sinusoids. The architecture has three bidirectional Transformer blocks: a Profile State Encoder (processes profile tokens with RoPE over log-seconds-to-life-long-event), an Event Encoder (processes each event independently, outputs per-event [EVT] tokens with calendar-feature augmentation), and a History Encoder (contextualises the concatenated [USR] + [EVT] sequence with RoPE over log-seconds-to-most-recent-event). Pre-training uses a masked-language-modelling objective at three granularities — token-level (15%), event-level (10%), and semantic-type/key-level (10%) masking — with label-smoothed cross-entropy, on 26M user records (24B events, 207B tokens) spanning 111 countries over 25 months (2023-2025). Training infrastructure uses dynamic batching, sequence packing with varlen attention, and two-level truncation (24 tokens/event, 6500 events/user). PRAGMA-S/M/L train on 16-32 H100 GPUs over 2 days to 2 weeks. Downstream adaptation uses either frozen embedding probes (linear/logistic, L-BFGS) or LoRA (rank 8, alpha 8, 2-4% parameter overhead). Across six internal banking tasks — credit scoring, communication engagement, external fraud, product recommendation, recurrent transactions, and lifetime value — PRAGMA with LoRA consistently outperforms task-specific baselines trained from scratch on raw event sequences: gains range from +1.8% (lifetime value ROC-AUC) to +130.2% (credit scoring PR-AUC). Scaling is task-dependent: rare-event discrimination tasks (credit, fraud, communications) benefit strongly from scale while saturating tasks (lifetime value, recurrent transactions) plateau at PRAGMA-S. A pre-training ablation (Table 3) confirms that LoRA fine-tuning of the pre-trained backbone matches or exceeds same-architecture training from scratch on every task (+10.3-54.1% at PRAGMA-L). The paper honestly reports a -47.1% failure on anti-money-laundering (AML), attributed to the model processing users in isolation while AML requires cross-record relational signals.

## Strengths
- **Scale and task breadth unprecedented for financial event FMs.** 26M users across 111 countries, 207B tokens, and six diverse downstream tasks (spanning credit risk, engagement prediction, fraud, recommendation, recurrence, and lifetime value) substantially exceed the scope of prior financial-sequence models like FinLangNet or nuFormer, which target one or two tasks on a single event source.
- **Thoughtful tokenisation design.** The key-value-time triple decomposition with type-specific value encoding (percentile buckets for numerics, single tokens for categoricals, BPE subwords for text) is well-motivated by the heterogeneity of banking events. The soft logarithmic time transform 8*ln(1+t/8) elegantly compresses long ranges while preserving local granularity, and cyclical calendar features handle day-of-week/month periodicity without learned parameters.
- **Clean architectural separation of concerns.** The three-encoder design (Profile State / Event / History) provides natural modularity: profile state encodes slow-moving context, the Event Encoder processes each event independently (enabling sequence packing), and the History Encoder contextualises across the full history. This decomposition is both computationally efficient and conceptually clean.
- **Pre-training ablation is convincing (Table 3).** Comparing LoRA fine-tuning of the pre-trained backbone against same-size training from scratch on identical downstream data shows consistent gains (+18.6% PR-AUC on credit, +13.0% on communication engagement, +10.3% on product recommendation), confirming that masked modelling recovers task-relevant structure that a single-task supervised loss does not.
- **LoRA vs. embedding-probe comparison (Table 1 right panel).** Showing that LoRA consistently outperforms frozen-embedding probes across all tasks and scales isolates the contribution of parameter-efficient adaptation from the representation itself, confirming gains come from joint fine-tuning rather than probe capacity.
- **Honest failure-mode reporting.** Disclosing the -47.1% AML result and attributing it to the model's inability to capture cross-record relational structure (money-laundering requires network-level features across users) demonstrates intellectual honesty and clearly delineates the paradigm's boundary.
- **Engineering contributions are reproducibly described.** Dynamic batching by event count, sequence packing with varlen attention (FlashAttention), two-level truncation (event-level at 24 tokens, user-level at 6500 events), and LMDB + Parquet storage are detailed in Appendix C with enough specificity to guide reimplementation.

## Weaknesses
- **All results are relative — no absolute metrics.** Due to commercial sensitivity, only percentage improvements over task-specific baselines are reported. This makes it impossible for readers to calibrate whether the absolute performance is production-grade, whether the baselines are weak, or how gains compare across tasks (e.g., +130.2% on credit PR-AUC could mean 0.01→0.023 or 0.10→0.23 — very different practical implications).
- **Single proprietary platform, no public-dataset evaluation.** The entire evaluation is on one fintech's internal data. No public financial-event dataset (PaySim, IBM synthetic transactions, Kaggle credit-card fraud, Berka Czech bank dataset) is tested, making the results unreproducible and impossible to compare against published baselines.
- **No comparison to other financial-sequence foundation models.** FinLangNet, nuFormer, TransactionGPT, and the recently-concurrent Braithwaite et al. (2025) "Your spending needs attention" are cited in related work but never compared against experimentally, even on their own reported tasks. The "task-specific baseline" comparator is described only as training the same architecture from scratch — not as the state of the art.
- **Temporal generalization beyond the 25-month window is untested.** Pre-training spans 2023-2025. Whether representations transfer to events from before 2023 (historical cold-start users) or after 2025 (future distribution shifts in product features, new event types) is not evaluated.
- **Encoder-only design limits downstream flexibility.** The architecture cannot perform generative tasks (synthetic transaction generation, next-event forecasting, sequence completion). This is a design choice rather than a flaw, but the paper does not discuss this limitation or why decoder/encoder-decoder alternatives were ruled out.
- **No representation analysis or interpretability.** There is no probing of what the model learns: no attention-pattern analysis, no embedding-space visualization, no feature-importance attribution on downstream tasks. Understanding whether the model relies on temporal patterns, event co-occurrence, or value magnitudes would strengthen the contribution.
- **AML failure is reported but not addressed.** The -47.1% drop is attributed to cross-record relational structure, but no mitigation is explored — not even a simple experiment with graph-level features concatenated to the PRAGMA embedding, or a discussion of how the architecture might be extended to handle multi-user contexts.
- **The six tasks are all from the same platform.** Cross-institution transfer (pre-train on Platform A, fine-tune on Platform B) is the natural generalization test for a "foundation model" but is entirely absent. Without this, PRAGMA is better characterized as a multi-task model for a single organization.
- **Masking strategy ablation is missing.** Three granularity levels (token 15%, event 10%, key 10%) are used jointly, but no ablation tests their individual or pairwise contributions. The reader cannot tell whether event-level or key-level masking (which are the novel additions over standard MLM) are load-bearing.
- **Downstream datasets share the pre-training corpus.** The paper states "downstream datasets are assembled following the same protocol as the pre-training corpus" — this raises a concern about whether downstream test-set users also appeared during pre-training (data leakage), though the evaluation-point temporal ordering may mitigate this. The paper should address this explicitly.

## Key Questions for Authors

1. **What are the datasets and what is their scale/complexity?** Are they publicly available or private? What are the absolute values of the metrics rather than the relative lifts? Without this, the evidence is not externally verifiable.

2. **No statistical significance or standard deviation on results.** The paper does not report multi-seed runs. Are these cherry-picked best results or representative of expected variance?

3. **Is the LoRA adapter doing the heavy lifting?**
   - How is hyperparameter tuning for LoRA adapters conducted?
   - If the pre-trained models have truly learned transferable representations, should they not perform comparably well out-of-the-box with simple linear probes? The gap between LoRA and embedding-probe variants is large (e.g., +10.3% vs +68.1% on product recommendation at PRAGMA-L).
   - If models require adapter fine-tuning to show gains, in what sense are they truly "foundational" versus simply good initializations for multi-task fine-tuning?